# Investigating the Interactions of the Cucumber Mosaic Virus 2b Protein with the Viral 1a Replicase Component and the Cellular RNA Silencing Factor Argonaute 1

**DOI:** 10.3390/v16050676

**Published:** 2024-04-25

**Authors:** Sam Crawshaw, Alex M. Murphy, Pamela J. E. Rowling, Daniel Nietlispach, Laura S. Itzhaki, John P. Carr

**Affiliations:** 1Department of Plant Sciences, University of Cambridge, Downing Street, Cambridge CB2 3EA, UK; sc945@cam.ac.uk (S.C.); amm1013@cam.ac.uk (A.M.M.); 2Department of Pharmacology, University of Cambridge, Tennis Court Rd., Cambridge CB2 1PD, UK; pjer2@cam.ac.uk (P.J.E.R.); lsi10@cam.ac.uk (L.S.I.); 3Department of Biochemistry, University of Cambridge, Sanger Building, 80 Tennis Court Rd., Cambridge CB2 1GA, UK; dn206@cam.ac.uk

**Keywords:** intrinsically disordered protein, protein phase separation, protein condensate, protein folding, RNA silencing suppressor regulation

## Abstract

The cucumber mosaic virus (CMV) 2b protein is a suppressor of plant defenses and a pathogenicity determinant. Amongst the 2b protein’s host targets is the RNA silencing factor Argonaute 1 (AGO1), which it binds to and inhibits. In *Arabidopsis thaliana*, if 2b-induced inhibition of AGO1 is too efficient, it induces reinforcement of antiviral silencing by AGO2 and triggers increased resistance against aphids, CMV’s insect vectors. These effects would be deleterious to CMV replication and transmission, respectively, but are moderated by the CMV 1a protein, which sequesters sufficient 2b protein molecules into P-bodies to prevent excessive inhibition of AGO1. Mutant 2b protein variants were generated, and red and green fluorescent protein fusions were used to investigate subcellular colocalization with AGO1 and the 1a protein. The effects of mutations on complex formation with the 1a protein and AGO1 were investigated using bimolecular fluorescence complementation and co-immunoprecipitation assays. Although we found that residues 56–60 influenced the 2b protein’s interactions with the 1a protein and AGO1, it appears unlikely that any single residue or sequence domain is solely responsible. In silico predictions of intrinsic disorder within the 2b protein secondary structure were supported by circular dichroism (CD) but not by nuclear magnetic resonance (NMR) spectroscopy. Intrinsic disorder provides a plausible model to explain the 2b protein’s ability to interact with AGO1, the 1a protein, and other factors. However, the reasons for the conflicting conclusions provided by CD and NMR must first be resolved.

## 1. Introduction

Cucumber mosaic virus (CMV) has a tripartite, single-stranded, positive-sense RNA genome [1,2]. RNA1 encodes the 110 kDa 1a protein, which functions as a methyltransferase and RNA helicase. RNA2 is the translation template for the 97 kDa 2a protein, which is an RNA-dependent RNA polymerase, and RNA2 also encodes the multifunctional 2b protein, which is translated from the RNA2-derived subgenomic RNA4A [1,2,3,4]. RNA3 is the translation template for the movement protein, while the coat protein is expressed from the RNA3-derived subgenomic RNA4 [1,2]. CMV has a very wide host range, comprising plants of 1071 species belonging to 521 genera [5], and the virus is transmitted in a stylet-borne, nonpersistent manner by aphid vectors from over 80 species [6,7,8].

Taxonomically, *Cucumber mosaic virus* is the type species of the *Cucumovirus* genus (family *Bromoviridae*) [5], which also contains the species *Gayfeather mild mottle virus*, *Peanut stunt virus*, and *Tomato aspermy virus* (TAV) [9]. However, CMV is the most diverse of the cucumoviruses. Most of the many strains and isolates of CMV can be assigned to one of three subgroups (i.e., Subgroups IA, IB, and II) based on RNA sequence similarity [1,2,10].

The multifunctional CMV 2b protein suppresses antiviral RNA silencing by binding to double-stranded short-interfering RNAs (siRNAs), and it also inhibits micro(mi)RNA-directed cleavage of host transcripts by Argonaute (AGO)1 and AGO4, AGO1-mediated regulation of mRNA translation, and AGO4-mediated effects on plant genomic DNA methylation (reviewed by Carr and Murphy [11]). The CMV 2b protein also inhibits plant defensive signaling pathways regulated by the phytohormones jasmonic acid and salicylic acid, and it influences interactions between host plants and the aphid vectors of CMV [11]. In *Arabidopsis thaliana*, the 2b protein’s effects on miRNA-directed AGO1 activity are mediated by a direct physical interaction [12,13]. The extent to which AGO1 activity is inhibited differs between 2b proteins encoded by CMV strains belonging to different subgroups, with 2b proteins of Subgroup II having a weaker inhibitory effect on miRNA-directed mRNA cleavage than orthologues encoded by most Subgroup IA and IB CMV strains [14]. However, recent work suggests that this is not due to an inability of Subgroup II CMV 2b protein orthologues to form 2b-AGO1 complexes; rather, it is due to differences in intracellular localization. Specifically, AGO1-2b complexes for Subgroup II orthologues (such as the 2b protein encoded by LS-CMV) accumulate almost exclusively in nuclei, but for Subgroup IA and IB 2b orthologues, these complexes also occur in the cytoplasm, consistent with the localization of the pool of AGO1 molecules mediating miRNA-mediated mRNA cleavage [15].

The more efficient inhibition of AGO1-mediated mRNA cleavage by 2b protein orthologues encoded by Subgroup IA or IB CMV strains helps explain why these strains generally induce more severe symptoms than strains belonging to Subgroup II [14,16,17]. However, unrestricted binding of the 2b protein to AGO1 can trigger effects that are directly or indirectly deleterious to the virus. For example, in *A. thaliana*, the inhibition of miR403-directed cleavage of *AGO2* mRNA permits increased AGO2 protein synthesis, which fosters increased RNA silencing-mediated resistance against CMV [18]. Additionally, inhibition of AGO1 activity by the 2b protein can induce strong resistance against aphid vectors. During infection, both effects are circumvented by the intervention of the CMV 1a protein, which binds to and re-localizes 2b protein molecules to processing bodies (P-bodies), which decreases the proportion of the 2b protein pool available for 2b-AGO1 complex formation [19].

Despite its important consequences for the success of CMV infection and for vector-mediated transmission of the virus, the interaction between the 2b protein and AGO1 is not understood in detail. For example, whereas specific amino acid residues involved in double-stranded RNA binding, RNA silencing suppression, or intracellular localization have been definitively characterized (reviewed in [11]), no specific 2b protein residues have been shown to be indispensable for the CMV 2b protein-AGO1 interaction to occur. Furthermore, it is unknown if specific 2b protein residue(s) are responsible for the interaction with the CMV 1a protein or if the same amino acids are involved in 2b-AGO1 complex formation.

Early studies that revealed, using site-directed mutagenesis, several functional domains in the 2b protein of the Subgroup IA strain Fny-CMV, including those responsible for nuclear localization, protein phosphorylation, RNA binding, and RNA silencing suppression, did not find a residue or domain for binding to AGO1 or to AGO4 [13,20] (Figure 1a). A subsequent study using the SD-CMV 2b protein employed a strategy of larger-scale deletions, which led to certain regions of the 2b protein being ruled out as being important in AGO1 binding, but it was inferred that the sequence(s) responsible for 2b-AGO1 complex formation most likely lay somewhere in a very broad region spanning residues 38 through 94 (Duan et al., 2012) [21] (Figure 1a). However, so far as we are aware, no specific AGO1-binding residues or discrete sequence domains have been pinpointed. In this study, we set out to identify sequences within the 2b protein that enable it to interact with AGO1 and that condition its regulation by the CMV 1a protein. In the process, we found that some of the 2b protein’s properties may potentially be explainable by the possession of intrinsically disordered regions.

## 2. Methods

### 2.1. Plant Growth and in Planta Transient Expression of Recombinant Proteins

*Nicotiana benthamiana* Domin. (accession RA-4 [25]) plants were grown from seed in a Conviron (Winnipeg, Manitoba, Canada) growth room at 22 °C, 60% relative humidity, 200 μmol.m^−2^.s^−1^ of photosynthetically active radiation, and 16 h light and 8 h dark. Cells of *Agrobacterium tumefaciens* (GV3101) carrying constructs for transient expression were incubated with shaking overnight at 28 °C in 50 mL of LB medium [26] containing appropriate antibiotics. Cultures were centrifuged at 5000× *g* for 15 min, pellets re-suspended and diluted to an OD_600_ of 0.5 in 10 mM MgCl_2_, 10 mM 4-morpholineethanesulfonic acid pH 5.6, and 100 μM acetosyringone, and infiltrated into the abaxial side of the third or fourth true leaves of four-week-old *N. benthamiana* plants using a needle-less syringe.

### 2.2. Cloning and Mutagenesis

Constructs used for the expression of fluorescently tagged CMV 1a (pMDC32 background for RFP or GFP fusions and pROK background for sYFP fusions) or AGO1 proteins (pSITE background for RFP or GFP fusions and pROK background for sYFP fusions) have been previously described [13,19]. The new constructs used in this study were derived from the Fny strain of CMV accession NC002035 [27]. Full-length and mutant versions of the Fny-CMV 2b sequence were incorporated into pSITE vectors [28,29] to yield constructs encoding CMV 2b fusions with red fluorescent protein (RFP), green fluorescent protein (GFP), and N- or C-terminal split yellow fluorescent protein (YFPn or YFPc) tags fused to the C-termini. Plasmids encoding fluorescently tagged mutant CMV 2b proteins were generated using Gateway^®^ cloning following the protocol provided (Invitrogen, Thermo-Fisher Scientific, Paisley, UK) with *att*B-flanked DNA fragments cloned into the pDONR221 donor plasmid, verified by sequencing before transfer into pSITE destination plasmids. The various *2b* mutants used in this study were constructed using the primers detailed in Appendix A. Truncated versions of the CMV 2b protein were generated using mutagenic primers containing attB sequences to yield PCR products compatible with Gateway^®^ cloning for insertion into destination vectors. In-frame deletions and substitutions were generated using a Q5 Site-Directed Mutagenesis Kit (New England Biolabs, Hitchin, UK) using the primers listed in Appendix A. Constructs were authenticated by automated Sanger sequencing [30,31] conducted by Source BioScience (Cambridge, UK).

### 2.3. RNA Silencing Suppression Assays and Fluorescence Imaging

For RNA silencing suppressor activity assays, *N. benthamiana* leaves were co-infiltrated with *A. tumefaciens* cells carrying plasmids expressing free GFP and cells carrying plasmids encoding wild-type or mutant CMV 2b proteins as described previously [19]. Leaves were imaged at 4, 8, and 12 days after agroinfiltration. For other imaging of fluorophores, imaging was at 4 days post-agroinfiltration. GFP, RFP, or reconstituted YFP fluorophores were imaged by laser excitation at 488, 561, or 514 nm, respectively, using a Leica Model SP5 confocal laser scanning microscope (Leica Microsystems, Heidelberg, Germany). The intensity of fluorescence was quantified using Image J. The R statistical package 3.2.2 (CRAN-Ma, Imperial College, London, UK, www.R-project.org: URL accessed on 3 November 2023) was used to perform an ANOVA and Tukey’s HSD *post hoc* test to assess the statistical significance of differences in intensity.

### 2.4. Protein Extraction, Coimmunoprecipitation, and Western Immunoblot Analysis

Total protein was extracted from 100 mg of agroinfiltrated *N. benthamiana* leaf tissue. Samples were frozen and pulverized in liquid nitrogen before homogenization in 25 mM Tris-HCl pH 7.5, 200 mM NaCl, 1 mM ethylenediaminotetraacetic acid (EDTA), 0.15% IGEPAL^®^ CA-630, 10% glycerol, 10 mM dithiothreitol, and protease inhibitor cocktail (Roche, West Sussex, UK). Crude extract was pelleted by centrifugation at 12,000× *g* for 4 min at 4 °C, and the clarified supernatant was collected. Coimmunoprecipitation of GFP- or RFP-tagged proteins from total leaf protein extract by incubation with GFP- or RFP-Trap magnetic agarose beads (ChromoTek, Planegg-Martinsried, Germany) was performed as previously reported [19]. Proteins were resolved on 10% polyacrylamide denaturing gels [32] and electrophoretically transferred onto a nitrocellulose membrane [33]. For immunological detection, membranes were probed with primary antibodies anti-GFP (1:1000) or anti-RFP (1:2000) (Chromotek) for 1 h, followed by incubation with horseradish peroxidase-conjugated secondary antibodies. Antibody binding was detected on X-ray film with Pierce ECL Substrate (Thermo-Fisher Scientific) or signals were directly captured in a G:BOX Chemi XRQ machine (Syngene, Cambridge, UK).

### 2.5. In Silico Analysis of Intrinsically Disordered Protein Sequences

IUPred3, ANCHOR2, and ParSe v2 prediction algorithms were used to assess the disordered nature of protein sequences for Fny-CMV 2b (accession: NC002035), LS-CMV 2b (accession: AF416900), Fny-CMV 1a (accession: D00356), LS-CMV 1a (accession: AF416899), and A. thaliana AGO1 (UID: 841262). The IUPred3 server [34] is a combined web interface that uses neural network strategies, educated with experimental data, to predict regions of disorder. The ANCHOR2 prediction algorithm [35] identifies context-dependent protein disorder, where the transition between the unstructured and structured states is initiated by the presence of an appropriate protein partner. ParSe v2 uses sequence hydrophobicity to identify intrinsically disordered protein sequences, followed by subsequent sorting into intrinsically disordered protein sequences that phase transition and those that do not [36]. The tertiary structure of the Fny-CMV 2b protein was also predicted using the online servers RoseTTAFold (https://robetta.bakerlab.org/, URL accessed on 3 November 2023; [37]); I-TASSER (https://zhanggroup.org/I-TASSER/, URL accessed on 30 October 2023; [38]); AlphaFold 2 (https://colab.research.google.com/github/sokrypton/ColabFold/blob/main/AlphaFold2.ipynb?fbclid=IwAR22AEhvCwA7VL4eEI6oMBGxITOSXyPTDZPNixZm5OQ59eDkpcbwJXeXipM#scrollTo=kOblAo-xetgx, URL accessed on 22 January 2024; [39])

### 2.6. Structural Analysis of the 2b Protein Using Nuclear Magnetic Resonance and Circular Dichroism

Synthesis of the Fny-CMV 2b protein was carried out by LifeTein protein expression services (Somerset, NJ, USA). His-tagged Fny-CMV 2b protein was expressed in *E. coli* under a T7 promoter and purified using Ni-NTA nickel affinity beads (Qiagen, Hilden, Germany). The purified protein was dissolved in phosphate-buffered saline and lyophilized. The 2b protein open reading frame (ORF) expressed matched the amino acid sequence of Fny-CMV 2b protein (accession NC002035) with a single amino acid substitution replacing glutamate at position 104 with lysine (E104G) to mitigate the toxic effects of the 2b protein in *E. coli* [24], yielding the following synthetic polypeptide as analyzed by mass spectrometry:

MELNVGAMTNVELQLARMVEAKKQRRRSHKQNRRERGHKSPSERARSNLRLFRFLPFYQVDGSELTGSCRHVNVAELPESEASRLELSAEDHDFDDTDWFAGNKWAEGAFLEHHHHHH.

The ^1^H NMR spectra were acquired at 25 °C using 600 mL of 200 µM protein in phosphate-buffered saline (137 mM NaCl, 2.7 mM KCl, 10 mM Na_2_HPO_4_, 1.8 mM KH_2_PO_4_, pH 7.4) amended with 1 mM DTT. The protein sample was subjected to 2-dimensional nuclear Overhauser enhancement spectroscopy (2D NOESY) (800 MHz, 298K, mixing time = 150 ms) and 2-dimensional total correlation spectroscopy (2D TOCSY) (800 MHz, 298K, mixing time = 32 ms) using an Avance III AV800 Spectrometer (Bruker, Billerica, MA, USA) equipped with a 5 mm TXI cryoprobe.

For CD analysis, a sample of the synthetic 2b protein was dissolved in 10 μM potassium phosphate, pH 7.4, with 1 mM 1,4-dithioerythritol using a PD MidiTrap™ G-25 buffer exchange column (Cytiva, UK). Far-UV CD spectra were obtained at 25 °C for a 13.5 mM sample of protein solution using a 1 mm path length quartz cuvette in a Chirascan CD spectrometer (Applied Photophysics, Leatherhead, UK). The CD spectrum was averaged across five scans recorded in the far-UV region (190–240 nm). CD data were analyzed using the K2D2 [40], CONTINSP175 [41], and CDSSTR [42] programs, which use reference data sets to predict protein secondary structures. A combination of different data sets was used, including a new reference dataset, IDP175, which is suitable for analyses of proteins containing significant proportions of disordered secondary structure [43].

## 3. Results

### 3.1. Construction of 2b Protein Deletion Mutants

To test the importance of different sequences of the Fny-CMV 2b protein in interactions with the CMV 1a protein and with AGO1, a series of cDNA clones encoding mutant versions of the protein were generated, in which regions with already known or currently unknown biological functions were deleted (Figure 1b). These included reconstruction of previously examined protein variants truncated in the N-terminal 17 (2b^Δ1–17^) or the C-terminal 16 (2b^Δ95–110^) amino acids [16], deletion of 55 N-terminal amino acids (2b^Δ1–55^), as well as mutants encoding 2b variants with more extensive deletions starting from the C-terminus. These were deletions of 26 (2b^Δ85–110^), 28 (2b^Δ83–110^), 37 (2b^Δ74–110^), 42 (2b^Δ69–110^), 46 (2b^Δ65–110^), 50 (2b^Δ61–110^), or 55 (2b^Δ56–110^) amino acids. There was a focus on deletions of the C-terminal/proximal sequences since we initially hypothesized that the C-terminal domain of the 2b protein might be involved in its interaction with the CMV 1a protein. The rationale for this was that the interaction between the CMV 1a and CMV 2b proteins ameliorates the 2b-induced symptom-like phenotype in transgenic plants [19,44] and that the 2b protein C-terminal domain is associated with symptom severity [16].

To investigate the potential roles of internal 2b protein sequences, in-frame internal deletions of residues 83 to 93 (2b^Δ83–93^), 56 to 65 (2b^Δ56–65^), 56 to 60 (2b^Δ56–60^), and 39 to 48 (2b^Δ39–48^) were constructed, as well as alanine substitution mutations replacing the native Fny-CMV 2b sequence at residues 56 through 65 with ten alanine residues (2b^56aaa65^) and 56 through 60 with five alanine residues (2b^56aaa60^). Lastly, two chimeric CMV 2b proteins were generated: one in which residues 83–93 of the Fny-CMV 2b sequence were introduced into the LS-CMV 2b sequence (2b^LS/Fny(83–93)^), and in the second two point mutations (Y58H and Q59G) into the Fny-CMV 2b protein sequence to recapitulate the LS-CMV 2b protein sequence between residues 56 and 60 (2b^Fny/LS(56–60)^) (Figure 1b).

It is possible that the perceived loss of interaction for some of the mutations in the CMV 2b protein may relate to a loss of stability in the protein. Additionally, some of the mutations generated in the 2b protein sequence disrupted its ability to act as a viral suppressor of RNA silencing (Appendix A). However, mutant proteins were visualized with RFP or GFP tags (Appendix A) and were all able to form homodimers with full-length Fny-CMV 2b protein molecules (Appendix A). This suggests that the tagged versions of the mutant proteins were sufficiently stable for co-localization studies and BiFC analysis.

### 3.2. Residues 56–60 of the Fny-CMV 2b Protein Are Required for Interaction with the CMV 1a Protein

Complex formation between the CMV 1a and 2b proteins causes the bound 2b protein molecules to be re-localized into P-bodies [19]. Monitoring whether this change in localization occurred was used as an initial means to assess whether mutant 2b proteins had lost the ability to interact with the 1a protein. Confocal laser scanning microscopy was used to examine the distribution of 2b-derived and 1a-derived C-terminal RFP and GFP fusion proteins (Figure 2 and Appendix A).

Deletion of the N-terminal 17 or C-terminal 16 residues did not abolish co-localization of the 2b protein with the 1a protein (Figure 2), although the C-terminal deletion caused an apparent decrease in the proportion of 2b protein co-localizing with the 1a protein (Appendix A). More extensive deletions (of residues 85 to 110 and 56 to 110, which bisected the 2b protein), as well as in-frame deletions of residues 83 to 93 and 56 to 60, resulted in the abolition of co-localization with the 1a protein (Figure 2). Replacement of residues 56–60 with alanine (mutant 2b^56aaa60^) also abolished co-localization with the CMV 1a protein, while replacement of residues 56–60 of the Fny-CMV 2b with the corresponding sequence from the LS-CMV 2b protein did not completely abolish co-localization with the CMV 1a protein. This was puzzling since the LS-CMV 2b protein is unable to interact with the Fny-CMV 1a protein [15].

The region between residues 83–93 in the Fny-CMV 2b protein has no equivalent sequence in the LS-CMV 2b protein (Figure 1 and Appendix A). Thus, we hypothesized that this sequence or residues lying within it might govern the interaction of the Fny-CMV 2b protein with the Fny-CMV 1a protein. However, the chimeric 2b^LS/Fny83–93^ protein created by insertion of these amino acids into the LS-CMV 2b protein backbone showed no co-localization with the Fny-CMV 1a protein (Figure 2). This indicates that the residues 83–93 of Fny-CMV 2b do not facilitate the 2b-1a interaction.

The effect of C-terminal deletions on 2b protein localization likely relates to a loss of a recently described nuclear export sequence (NES) between residues 77–87 (Figure 1). The deletion of the N-terminal residues 1–17 did not appear to impact nuclear localization, but deletion of residues from 1–55, which includes the NLS domain, caused a reduction in nuclear localization of the 2b protein (Appendix A). The results are confirmatory of previous findings with the Fny-CMV 2b protein [16].

To authenticate that observed decreases in, or abolition of, co-localization between 2b-derived and 1a-derived C-terminal RFP and GFP fusion proteins genuinely reflected genuine losses of direct interactions, bimolecular fluorescence complementation (BiFC) assays were carried out. To facilitate this, the N- and C-terminal domains of the yellow fluorescent protein (YFPn or YFPc) were fused to the C-termini of mutant CMV 2b proteins and the C-terminus of the Fny-CMV 1a protein. In contrast to the co-localization results obtained with the deletion mutant 2b^Δ83–93^ protein but in agreement with the co-localization results obtained with the chimeric 2b^LS/Fny83–93^ protein (Figure 2), deletion of residues 83–93 did not abolish the physical interaction between the 2b and 1a proteins; however, more extensive deletions of residues in the C-terminus did (Figure 3 and Appendix A).

According to BiFC assays, deletion of the 2b protein residues from 61 or 65 to the C-terminal residue 110 diminished the interaction with the CMV 1a protein, and deletion of residues from 56 to 110 abolished the interaction completely (Figure 3). In-frame deletions of residues 56–65 or 56–60 also abolished 1a-2b protein complex formation, as did substitution of the authentic amino acids at positions 56–65 and 56–60 with alanine residues (respectively, mutants 2b^56aaa65^ and 2b^56aaa60^: Figure 1b) (Figure 3). Mutation of the sequence 56 to 60 of the Fny-CMV 2b protein (PF*YQ*V) to recapitulate the corresponding sequence of the LS-CMV 2b protein (PF*HG*V) did not completely abolish the 2b-1a protein–protein interaction, although it appeared to be weaker (Figure 3), suggesting that while this domain is the major determinant of binding, other Fny-CMV 2b sequences may play some role.

Co-immunoprecipitation assays confirmed the importance of residues 56–60 of the Fny-CMV 2b protein in mediating complex formation with the 1a protein (Figure 4). In agroinfiltrated leaves of *N. benthamiana*, GFP-tagged Fny-CMV 1a protein was co-expressed with RFP-tagged Fny-CMV 2b protein mutants. RFP-tagged wild-type and mutant 2b proteins were immunoprecipitated from leaf homogenates using magnetic agarose beads coated with anti-RFP and analyzed by immunoblotting using anti-GFP to detect Fny-CMV 1a protein molecules complexed with 2b-RFP fusion proteins. The Fny-CMV 1a protein-GFP fusion co-immunoprecipitated with the wild-type 2b-RFP fusion protein and the RFP-tagged variant of the chimeric 2b^LS/Fny56–60^ protein but not with the RFP fusion protein variants of the 2b^Δ56–60^ or 2b^56aaa60^ mutants (Figure 4), in line with results from the BiFC assays (Figure 3).

### 3.3. Residues 56–60 Are Important for the Interaction of the Fny-CMV 2b Protein with AGO1 as Well as with the CMV 1a Protein

The effects of mutations in the CMV 2b protein on its interactions with AGO1 were examined. Deletions of residues 1–17 or 83–93 had no impact on the co-localization of AGO1 with the 2b protein (Figure 5). However, progressively larger deletions of residues beginning at the C-terminus (95–110, 85–110, 74–110, 56–110, and 48–110) diminished AGO1-2b protein co-localization (Figure 5). To determine if these changes in localization resulted from losses of physical interaction between the two proteins, BiFC was carried out (Figure 6). Deletion of residues 1–17, 95–110, 85–110, or 74–110 had no impact on the interaction of the 2b protein with AGO1. However, marked decreases in the ability of the 2b protein to interact with AGO1 occurred after deletion of residues 61–110, 56–110, or 48–110. In-frame deletions of residues 56–60 or 39–48 weakened the interaction with AGO1, as did replacement of amino acids in the Fny-CMV 2b protein to match residues 56–60 of the LS-CMV 2b protein (the chimeric 2b^LS/Fny56–60^ protein). Substitution of residues 56–60 with alanine residues (mutants 2b^56aaa60^) also decreased interaction with AGO1.

Interestingly, the CMV 2b mutant protein lacking residues between 1–55 still interacted with AGO1, but its localization was predominantly cytoplasmic, likely due to the loss of the NLS region. Thus, both C- and N-proximal 2b protein sequences may facilitate the interaction with AGO1. Our evidence for the importance of sequences between residues 39 and 60 in the Fny 2b sequence is not inconsistent with previous work [21], but it throws doubt on whether there is a single discrete region of the 2b protein that is alone sufficient for AGO1 interaction.

Co-immunoprecipitation assays confirmed the importance of residues 56–60 in mediating the interaction of the Fny-CMV 2b and AGO1 proteins (Figure 7). RFP-tagged mutant versions of the Fny-CMV 2b protein were co-expressed with GFP-tagged AGO1 protein, immunoprecipitated using anti-RFP agarose magnetic beads, and analyzed by western immunoblotting using anti-GFP antibodies to detect any AGO1 proteins complexed with the mutant 2b-RFP proteins. The AGO1-GFP fusion co-immunoprecipitated with the full-length 2b-RFP protein and the chimeric 2b^LS/Fny56–60^-RFP protein but not with the 2b^Δ56–60^-RFP or 2b^56aaa60^-RFP proteins (Figure 7), in line with results from the BiFC assays.

### 3.4. In Silico Prediction of Intrinsic Disorder in the 2b Protein

The importance of residues 56–60 in the 1a-2b interaction and residues 39–60 in the AGO1–2b interaction (which overlap with the sequence needed for 1a-2b complex formation) suggests that larger regions of the 2b protein structure are required for these interactions. Such effects can occur when two or more proteins with intrinsically disordered structures interact [45] and may also explain 1a-2b protein complex formation in P-bodies [46].

The IUPred3 [34] and ParSe v2 [36] programs were applied to the full-length sequence of the Fny-CMV 2b protein. Both systems predicted that the Fny-CMV 2b protein contains two intrinsically disordered regions between residues 20–44 and 70–88 (Figure 8). Additionally, the ANCHOR2 algorithm [35] predicted the presence of two regions capable of binding another molecule, thereby forming a more ordered structure (Figure 8a). Predictions using the CIDER server (http://pappulab.wustl.edu/CIDER/about/: URL accessed on 18 April 2024) suggest that the 2b protein has a ‘Janus’ region with a highly context-dependent structure (Appendix A). Interestingly, the 2b protein of the mild CMV strain LS-CMV (Subgroup II) was predicted to have a more ordered structure with only one potential region of intrinsic disorder between residues 28–38 (Figure 8b). The 2b protein orthologues encoded by Subgroup IB CMV strains showed an intermediate level of disorder in their structure, with the same predicted region of disorder between residues 20–44 but a smaller disordered region between residues 83–90 and only one predicted disordered binding domain between residues 1–20 (Figure 8b). The disordered nature of the CMV 1a and AGO1 proteins was also predicted using the same algorithms. These results indicate that the 1a protein from Fny-CMV contains a region of disorder between residues 539–565 and LS 2b contains a much smaller region of disorder between residues 556–565. AGO1 also contains disordered regions at its N- and C-termini (as has previously been predicted: [47]).

A common feature of intrinsically disordered proteins is their ability to undergo phase separation. The key determinant of whether an intrinsically disordered protein will phase transition is the balance between residues driving cohesive intramolecular interactions and the polar residues driving solvent interactions [48,49,50]. There have been several methods developed to predict sequences that drive separation [51,52]. One such tool is ParSe v2, which uses sequence hydrophobicity to identify intrinsically disordered protein regions, followed by subsequent sorting into intrinsically disordered protein regions that phase transition and those that do not [36]. ParSe v2 predicted that both CMV 1a and 2b proteins contain residues promoting phase separation (Appendix A). Similarly, the AGO1 protein contains residues that promote its phase separation (Appendix A).

### 3.5. Circular Dichroism and Nuclear Magnetic Resonance Results Differ with Respect to the Disorderedness of the 2b Protein

Circular dichroism (CD) was used to explore secondary structure composition. The results suggested that more than 50% of the Fny-CMV 2b protein is intrinsically disordered. The CDSSTR program [53] was used, and the best fit of the data predicted that the protein was 13% α-helix, 32% β-sheet, and 55% disordered (Figure 9, Appendix A). The indication of a large component of β-sheet from the CD analysis was an unexpected finding since it had been assumed that the 2b protein is mainly α-helical. This assumption was based on X-ray crystallography results for the resolvable portion of the orthologous TAV 2b protein and on modeling by Gellèrt et al. [54]. However, the TAV 2b protein shares only 58% sequence identity with the CMV 2b protein (Appendix A), and structural studies for the CMV 2b protein have not been conducted before. Folding predictions also suggest a largely α-helical N-terminal half of the protein with no β-sheet structure present (Appendix A).

However, the results from ^1^H nuclear magnetic resonance (NMR) spectroscopy indicated that a high proportion of the Fny-CMV 2b protein adopts α-helical structure, with other regions forming β-sheets. The spectra are consistent with a typical globular protein with at least 40 residues arranged in α-helices. There was also evidence for at least 20 residues forming a β-sheet region (Figure 10). In contrast to the secondary structure composition indicated by CD, NMR detected few signals indicative of intrinsically disordered regions within the Fny-CMV 2b protein (Figure 10).

## 4. Discussion

### 4.1. Controlling Interactions of the CMV 2b Protein with the CMV 1a Protein and AGO1

The CMV 2b protein interaction with AGO1 can trigger AGO2-mediated antiviral RNA silencing against CMV [18] as well as a form of resistance against its aphid vectors [44]. Presumably, inhibition of AGO1 is somehow still advantageous for the virus, despite these potentially deleterious effects, which may explain why the ability of 2b to complex with AGO1 has been conserved and why a mechanism has evolved by which the 1a protein interacts with the 2b protein sufficiently to diminish but not abolish its effects on AGO1. Meanwhile, and despite the likely importance of the interaction, no specific amino acids have been identified as being responsible for the 2b protein’s interaction with AGO1 (or with AGO4). Generally, only broad inferences have been made regarding the central region of the 2b protein, suggesting that it may be involved [13,21,55]. In some ways, this is puzzling since, despite the smallness of the 2b protein, many specific residues and sequence domains have been identified that control other activities carried out by this highly multifunctional protein. A potential conclusion is that it may not be possible to pinpoint a single domain of the CMV 2b protein that governs its interaction with AGO1, and that multiple sequences or larger regions of the protein may be involved. A possibility that our work raises is that 2b-AGO1 complex formation may be controlled by intrinsically disordered regions within the 2b protein’s secondary structure. If correct, this might help explain aspects of 2b-1a complex formation and why so many other interactors have been described for the CMV 2b protein.

### 4.2. Residues 56 to 60 of the CMV 2b Protein Are Involved in Its Interaction with Both the CMV 1a Protein and AGO1

With respect to complex formation between the 1a and 2b proteins of CMV, since the N- and C-terminal domains of the 2b protein have been implicated in enhancement and inhibition of symptoms, respectively [16,56], we hypothesized that the C-terminal domain may play a role in mediating the interaction of the CMV 2b protein with the CMV 1a protein. However, while deletion of the C-terminal domain inhibited the interaction of the 2b protein with the 1a protein, it did not abolish it. Similarly, mutation of residues 83–93 decreased the strength of interaction between the CMV 1a and 2b proteins but did not abolish it. Our results suggest that residues 56–60 are necessary for interaction between CMV 1a and 2b proteins to occur. Mutation of these residues also weakened interactions with AGO1, and perhaps this overlap may contribute to the competition between AGO1 and 1a for binding to the Fny-CMV 2b protein [19]. However, it was not possible to attribute AGO-2b interactions to a discrete amino acid sequence within the 2b protein, and this prompted us to investigate if the folding of the CMV 2b protein rather than its primary sequence might condition its interactions with other proteins.

### 4.3. Modeling the Secondary Structure of the Fny-CMV 2b Protein

To our knowledge, it has not been possible to date to determine the structure of a CMV 2b protein using X-ray crystallography. However, a crystal structure has been determined to a resolution of 2.82 Å for a dimer of the orthologous 2b protein of the cucumovirus TAV complexed with a 19-base-pair double-stranded RNA molecule [57]. Unfortunately, this crystallographic structure is incomplete. Absent from the model are regions of TAV 2b, including its N-terminal four amino acids and its C-terminal residues, which the authors considered to be unstructured [57]. At the present time, the most detailed published insights into the three-dimensional structures of the CMV 2b protein have come from modeling; in particular, a model developed by Gellèrt et al. [54] predicted that the C-terminal domain of the CMV 2b protein is potentially unstructured with only a single short stable α-helix between residues 68–76.

Intrinsically disordered proteins are characterized by the absence of stable structures, and they exist instead as conformational ensembles [58,59]. Our modeling suggested there are marked differences in the numbers of disorder-promoting residues between 2b orthologues of Subgroup IA compared with Subgroup II CMV strains. The 2b ORFs of Subgroup IA CMV strains typically encode products 10 residues longer than Subgroup II CMV strains. Our alignment of the Fny-CMV (Subgroup IA) and LS-CMV (Subgroup II) 2b proteins placed this ‘missing’ sequence between residues 83 and 93 (based on the Fny-CMV 2b protein coordinates). However, there was a prediction of intrinsic disorder over a larger region of dissimilarity between residues 62 and 93 in 2b proteins encoded by Subgroup IA and IB CMV strains. In addition to modeling predictions, a number of observations point towards the disordered nature of the Fny-CMV 2b protein, such as the difficulties of obtaining X-ray crystallographic structures, toxicity following overexpression of the full-length 2b protein in bacteria [24,54], and differences between the predicted mass of the 2b protein and its apparent mass according to sodium dodecyl-sulfate polyacrylamide gel electrophoresis, all hallmarks of intrinsically disordered proteins [60,61,62,63]. The possibility of intrinsic disorder in the Fny-CMV 2b protein led us to experimentally investigate its secondary structure.

### 4.4. Experimental Investigations of the Folding of the 2b Protein

To our knowledge, the CD and NMR data presented here are the first obtained for the CMV 2b protein. The CD data broadly supported the modeling predictions that the C-terminal 50% of the protein is largely disordered. The CD analysis also suggests there are both α-helical structures and β-sheet structures, but surprisingly, the β-sheet content appeared to be greater than suggested by either our modeling or that of Gellèrt and colleagues [54]. The NMR data suggested the Fny-CMV 2b protein is mainly α-helical, which is in line with the structure predictions based on the crystallography-derived structure for TAV 2b [57]. Contrastingly, we found less evidence for the intrinsically disordered nature of the Fny-CMV 2b protein from our NMR results, which conflicted with modeling predictions as well as with the CD data. The NMR analysis also indicates a greater abundance of β-sheet than we had anticipated from our modeling, the previous modeling of the CMV 2b proteins such as that of Gellèrt et al. [54], or the partly solved X-ray crystallographic structure of the TAV 2b protein [57].

### 4.5. New Questions on the CMV 2b Protein Secondary Structure and Its Functional Effects

A possible explanation for the differing conclusions reached with NMR compared to those reached using computational approaches and CD measurements could be that NMR was carried out with 2b protein in phosphate buffered saline (containing 137 mM NaCl and 2.7 mM KCl), whereas CD was conducted in the absence of salt. The diagram of states generated as part of our modeling indicated that the Fny-CMV 2b sequence is a ‘Janus’ protein, i.e., possessing a highly context-dependent secondary structure that can dramatically and reversibly adopt different conformations in response to factors in its environment, including, *inter alia*, salt concentration. Interestingly, Gellèrt et al. [54] proposed that the interactions of metal ions with negatively charged residues in the 2b protein C-terminal domain would participate in β-sheet formation. The 2b protein is known to dimerize and form higher-order assemblies with itself [13,23,57]. Although the mechanism has not been investigated, it cannot be based on the formation of disulfide bridges, and another explanation is required. This could be the emergence of structure from the interactions of intrinsically disordered domains within the interaction partners [45,64,65,66,67,68]. It is plausible that in low-salt environments, the Fny-CMV 2b protein is monomeric, not interacting with positive ions, and forms a more unstructured state, while in higher-salt environments, the Fny-CMV-2b protein forms a more structured state due to self-interaction.

### 4.6. An Intrinsically Disordered Structure: An Intriguing Possibility That May Explain Several Properties of the CMV 2b Protein

At this point, we cannot be certain that the 2b protein possesses intrinsically disordered regions and, if it does, under what conditions the disorder will manifest itself. Further investigation of this possibility is warranted since it could explain several of its properties, in particular those that cannot be attributed to discrete residues or domains. For example, a key feature of intrinsically disordered proteins is their ability to interact with multiple partners [45,69]. This multivalency could explain variously reported interactions (in addition to those explored in this paper) with: DNA [24,70], a catalase [71] and a calmodulin-like factor in tobacco [72], cucumber RNA-directed RNA polymerase 1 [73], an *A. thaliana* zinc finger protein HB27 [74], and *A. thaliana* jasmonate ZIM-domain (JAZ) proteins 1, 3, 6, and 10 [75]. Intrinsically disordered proteins, including at least one plant viral protein (the p26 movement protein of pea enation mosaic virus 2), can phase separate and enter membrane-free organelles composed of protein-rich droplets [46,76,77,78,79]. The 2b protein of Fny-CMV localizes to two membrane-free organelles, P-bodies [19] and nucleoli [13], indicating it can phase-separate. Nevertheless, however plausible it may seem that CMV 2b is functioning as an intrinsically disordered protein, further work using a wider range of experimental conditions will be required to confirm this.

## Figures and Tables

**Figure 1 viruses-16-00676-f001:**
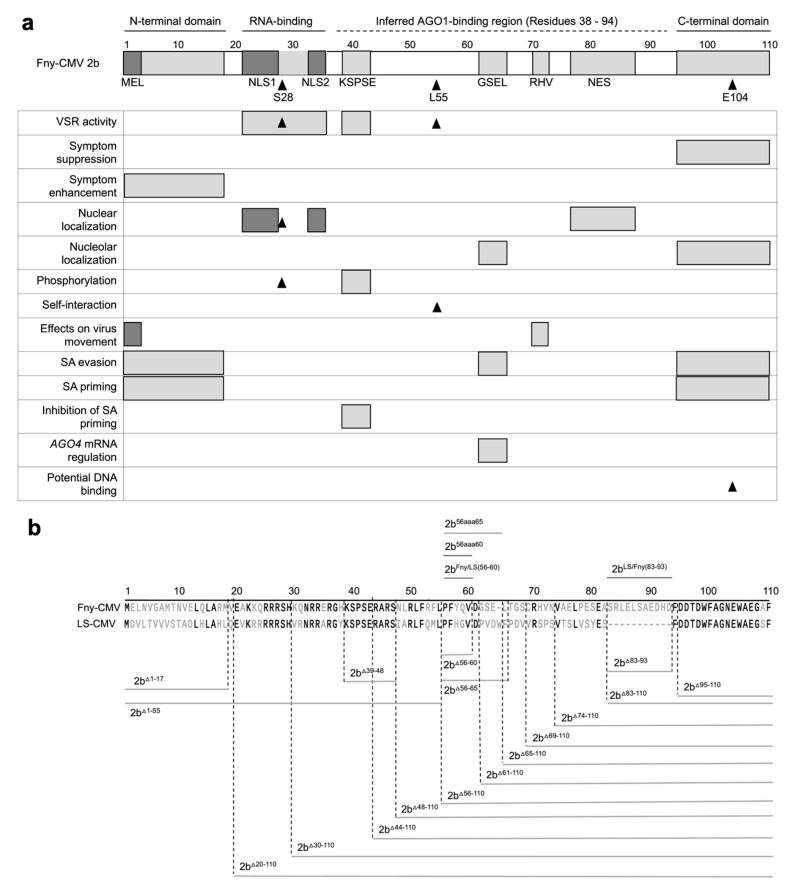
Mutational analysis of the cucumber mosaic virus (CMV) 2b protein. (**a**) Previously determined or inferred functional residues or domains of the 2b protein are indicated on a map of the 110 amino acid Fny-CMV 2b orthologue. Sequences longer than one amino acid are depicted as gray boxes: the N- and C-terminal domains; the RNA-binding domain; the KSPSE phosphorylation sequence; the GSEL and RHV sequences; and the nuclear export sequence (NES). However, the N-terminal MEL sequence and nuclear localization sequence (NLS) 1 and 2 are shaded in darker gray to indicate that they overlap with the N-terminal and RNA-binding domains, respectively. Single amino acid residues with known biological effects are indicated by arrowheads: S28, which is an additional phosphorylation site [22]; L55, which is required for 2b self-interaction [23]; and E104, which causes cytotoxicity in *E. coli* due to DNA binding [24], suggesting a biological function. The biological roles of sequences or residues are indicated on the left. The region spanning residues 38 to 94 of the 2b protein, inferred to contain amino acid residue(s) required for interaction with Argonaute 1 (AGO1), is indicated by a dashed line and was proposed by Duan et al. [21]. The map is updated from a previous iteration [11] to include more recent information [22]. (**b**) Deletion mutations are indicated by gray lines below the amino acid sequences for the Fny-CMV and LS-CMV 2b proteins, while insertions or substitution mutations are indicated above. Numbers refer to the residues of the 110 amino acids in the Fny-CMV 2b protein. Mutant 2b proteins were truncated from the N-terminus of the first 17 (2b^Δ1–17^) or 55 (2b^Δ1–55^) amino acids. Truncations were also made from the C-terminus, deleting 16 (2b^Δ95–110^), 26 (2b^Δ85–110^), 28 (2b^Δ83–110^), 37 (2b^Δ74–110^), 42 (2b^Δ69–110^), 46 (2b^Δ65–110^), 50 (2b^Δ61–110^), and 55 (2b^Δ56–110^) amino acids. In-frame internal deletions were also made between residues 83 to 93 (2b^Δ83–93^), 56 to 65 (2b^Δ56–65^), 56 to 60 (2b^Δ56–60^), and 39 to 48 (2b^Δ39–48^). Alanine substitution mutations were also made to replace the Fny-CMV 2b sequence from residues 56 to 65 with ten alanine residues (2b^56aaa65^) and residues 56 to 60 with five alanine residues (2b^56aaa60^). Two chimeric 2b proteins were generated. In 2b^Fny/LS(56–60)^, the mutations Y58H and Q59G were introduced into the Fny-CMV 2b sequence to recapitulate the sequence of the LS-CMV 2b orthologue between residues 56 and 60. The chimeric protein 2b^LS/Fny(83–93)^ was created by introducing residues 83–93 from the Fny-CMV 2b protein into the LS-CMV 2b protein background. Note that the wild-type LS-CMV 2b protein does not contain a corresponding sequence.

**Figure 2 viruses-16-00676-f002:**
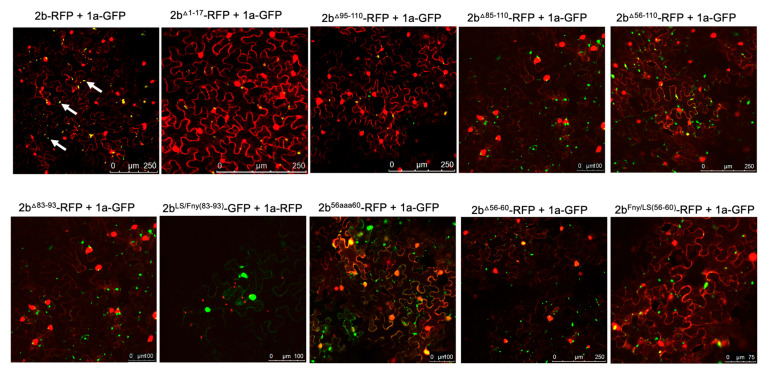
Subcellular localization of mutant 2b proteins and the full-length 1a protein. Using agroinfiltration, C-terminal RFP or GFP fusion proteins derived from the full-length 2b protein (2b-RFP) or mutant 2b proteins lacking residues 1–17, 95–110, 85–110, 56–110, 83–93, 56–60, or substitutions LS/Fny (83–93), 56aaa60, or Fny/LS (56–60) were co-expressed with 1a-RFP or 1a-GFP fusion proteins in *N. benthamiana* leaves in the combinations shown. Fluorescent signals were imaged using confocal scanning laser microscopy. Fny 2b-RFP accumulated in the nucleus and cytoplasm, with a proportion co-localizing with the 1a-GFP (merged signal shown as yellow and typical examples indicated with white arrows in the top left image), consistent with previous results [19]. Deletion of the N-terminal 17 residues (1–17) in the 2b sequence or the 15 C-terminal residues (95–110) did not disrupt the co-localization of 2b and 1a proteins. Deletions from 85–110 and 56–110 resulted in an apparent loss of co-localization with the 1a protein. Smaller in-frame deletions of residues 83–93 and 56–60 also both resulted in an apparent loss of co-localization between mutant 2b proteins and the 1a protein.

**Figure 3 viruses-16-00676-f003:**
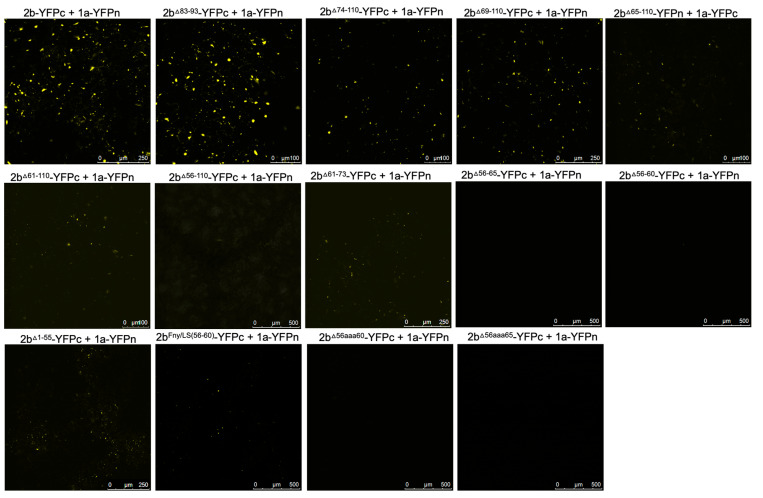
Interactions of mutant versions of Fny-CMV 2b protein with the 1a protein. Mutant versions of the 2b protein were fused at their C-termini with the C-terminal domain of the split yellow fluorescent protein (YFPc). Using agroinfiltration in *N. benthamiana* leaves, these fusion proteins were co-expressed with YFP N-proximal domain fusion proteins (YFPn) and the 1a protein of Fny-CMV. Direct protein–protein interactions in vivo were revealed by bimolecular fluorescence complementation and the resulting fluorescence imaged by confocal laser scanning microscopy. The images show that the 2b mutants lacking residues 83–93, 74–110, and 69–110 retained their ability to interact with the Fny-CMV 1a protein. Truncations of the 2b protein sequence between residues 65–110 and 61–110 resulted in progressively fewer instances of interaction with the 1a protein, and deletion of residues 56–110 abolished the interaction completely. In-frame deletions between residues 56–65 or 56–60 or alanine substitutions between these residues also resulted in no interaction between 2b and 1a proteins. However, substitution of the Fny-CMV sequence with that of LS-CMV between residues 56–60 did not abolish the interaction with the 1a protein (although the amount of 2b-1a interaction was seemingly decreased).

**Figure 4 viruses-16-00676-f004:**
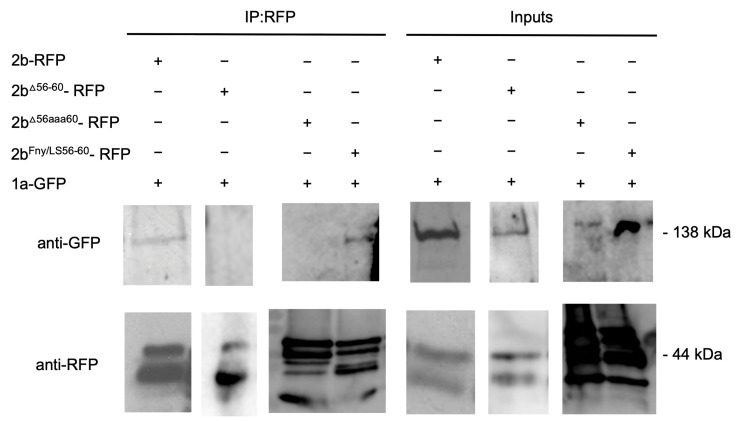
Interactions of the cucumber mosaic virus 1a protein with mutant versions of the CMV 2b protein in plants were examined by co-immunoprecipitation. Using agroinfiltration into *N. benthamiana* leaves, a GFP fusion protein derived from CMV 1a was co-expressed with RFP fusion proteins derived from the full-length 2b protein (2b-RFP) or mutant 2b sequences with deletions between residues 56–60 (2b^Δ56–60^-RFP), alanine substitutions between residues 56–60 (2b^56aaa60^-RFP), or replacement of the Fny-CMV 2b sequence with that of the LS-CMV 2b sequence between residues 56 and 60 (2b^Fny/LS(56–60)^-RFP). Total protein was extracted from leaf samples and immunoprecipitated with RFP-Trap beads (IP:RFP), followed by immunoblot analysis with anti-GFP antibodies to detect the 1a-GFP fusion protein. 1a-GFP was detected in all input samples with a corresponding band of approximately 138 kDa. However, following RFP-pulldown, 1a-GFP could only be detected when co-expressed with 2b-RFP or 2b^Fny/LS(56–60)^-RFP and not with the 2b^Δ56–60^-RFP or 2b^56aaa60^-RFP mutants. The original blots used to make the composite image are shown in Appendix A.

**Figure 5 viruses-16-00676-f005:**
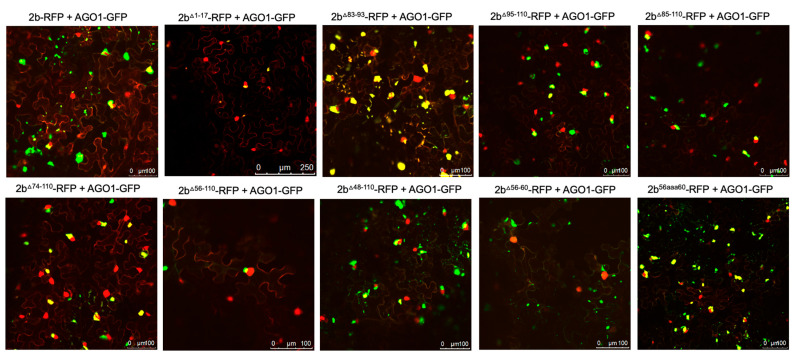
Subcellular localization of mutant 2b proteins and the Argonaute 1 protein (AGO1). Using agroinfiltration, C-terminal RFP or GFP fusion proteins derived from full-length 2b proteins or mutant 2b proteins were co-expressed with the AGO1-GFP fusion protein in *N. benthamiana* leaves in the combinations shown. Fluorescent signals were imaged using confocal scanning laser microscopy. Full-length Fny-CMV 2b-RFP protein accumulated in the nucleus and cytoplasm, with a proportion co-localizing with the AGO1-GFP (merged signal shown as yellow). Deletions of residues 1–17 or 83–93 had no impact on the co-localization of AGO1 and 2b signals. However, deletion of the C-terminal residues 95–110, 85–110, or 74–110 resulted in a reduction in the proportion of co-localized signal. Larger deletions in the 2b sequence between residues 56–110 and 48–110 resulted in noticeably reduced co-localization between mutant 2b proteins and AGO1, but co-localization was never completely abolished.

**Figure 6 viruses-16-00676-f006:**
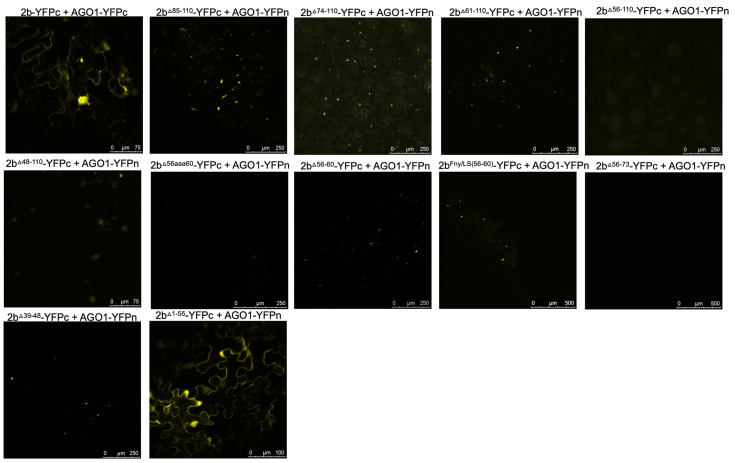
Interactions of wild-type or mutant versions of the Fny-CMV 2b protein with Argonaute 1 (AGO1). Mutant versions of the 2b protein were fused at their C-termini with the C-terminal domain of the split yellow fluorescent protein (YFPc). Using agroinfiltration in *N. benthamiana* leaves, these fusion proteins were co-expressed with YFP N-proximal domain fusion proteins (YFPn) and AGO1 proteins. Direct protein–protein interactions were revealed in vivo by bimolecular fluorescence complementation and the resulting fluorescence imaged by confocal laser scanning microscopy. The images show that deletion of residues 85–110 or 74–110 had no impact on the interaction of mutant 2b proteins with AGO1. Deletion of residues from 61–110 caused a noticeable decrease in interaction between 2b and AGO1 proteins, and deletions between 56–110 or 48–110 caused an almost complete loss of interaction. In-frame deletions between residues 56–60 or 39–48 resulted in a greatly weakened interaction with AGO1. Mutation of a 56–60 sequence to match that of the LS protein also resulted in a decreased strength of interaction. Alanine substitutions between residues 56–60 also resulted in decreased interaction with AGO1. However, mutant 2b proteins lacking residues between 1–55 were still able to interact with AGO1 and had a predominantly cytoplasmic localization, likely as a result of the loss of NLS sequences.

**Figure 7 viruses-16-00676-f007:**
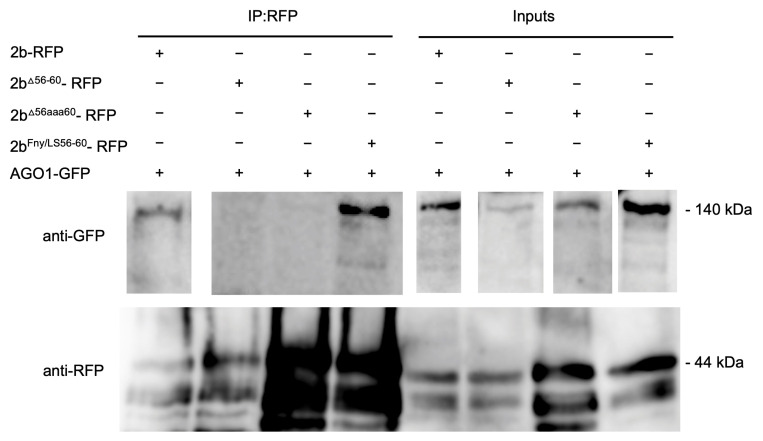
Co-immunoprecipitation showing that the mutant CMV 2b protein lacking residues 56–60 does not interact with AGO1. Using agroinfiltration into *N. benthamiana* leaves, a GFP fusion protein derived from AGO1 was co-expressed with RFP fusion proteins derived from full-length 2b proteins (2b-RFP) or mutant 2b sequences with deletions between residues 56–60 (2b^Δ56–60^-RFP), alanine substitutions between residues 56–60 (2b^56aaa60^-RFP), or replacement of the Fny-CMV 2b sequence with that of the LS-CMV 2b sequence between residues 56 and 60 (2b^Fny/LS(56–60)^-RFP). Total protein was extracted from leaf samples and immunoprecipitated with RFP-Trap beads (IP:RFP), followed by immunoblot analysis with anti-GFP antibodies to detect AGO1-GFP. AGO1-GFP was detected in all input samples with a corresponding band of approximately 140 kDa. However, following RFP-pulldown, AGO1-GFP could only be detected when co-expressed with 2b-RFP or 2b^Fny/LS(56–60)^-RFP and not with the 2b^Δ56–60^-RFP or 2b^56aaa60^-RFP. The original blots used to make the composite image are shown in Appendix A.

**Figure 8 viruses-16-00676-f008:**
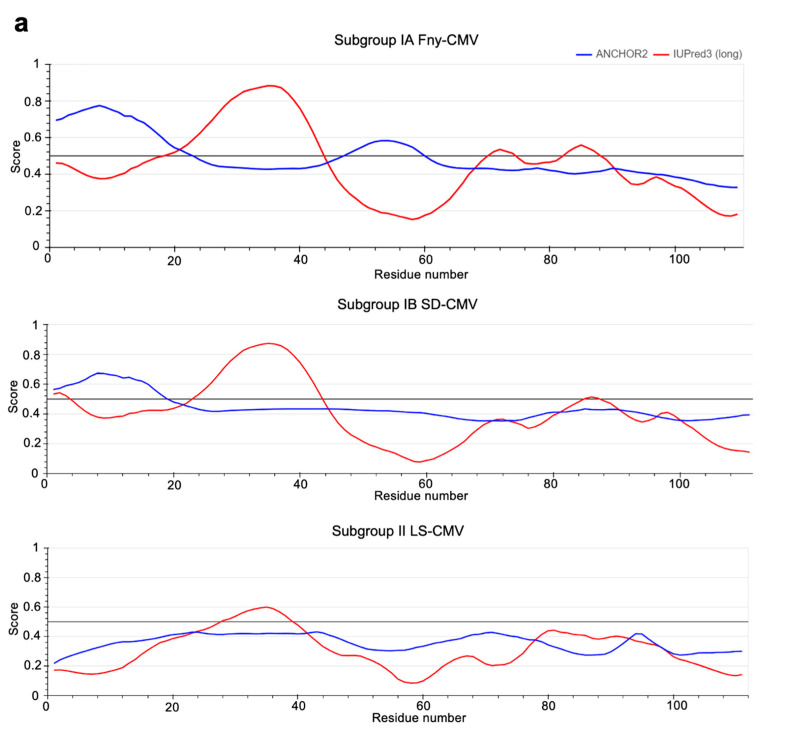
Predictions of intrinsically disordered regions. (**a**) The IUPred3 program (shown in red) predicted that approximately 50% of the Fny-CMV 2b protein is likely to be intrinsically disordered with two principal disordered regions between residues 20–44 and 70–88. The ANCHOR2 algorithm (shown in blue) predicted the presence of two regions capable of binding another molecule and forming an ordered structure between residues 1–20 and residues 48–60. The 2b proteins from SD-CMV (a Subgroup IB strain) had only one predicted region of disorder (between residues 20–44) but no disordered region between residues 83–90 and only one predicted disordered binding domain between residues 1–20. The 2b protein of LS-CMV was predicted to have a much more ordered structure with only one potential region of disorder between residues 28–38 and no predicted disordered binding domains. (**b**) The ParSe v2 program yielded similar results, with a greater proportion of the 2b protein predicted to be intrinsically disordered in Subgroup IA strains than Subgroup IB strains and no sequences with 20 or more contiguous residues that are at least 90% disorder-promoting found in 2b proteins encoded by Subgroup II strains. Protein sequences are depicted as horizontal bars with folded regions shaded black, intrinsically disordered regions shaded red, and mixed regions shaded white. The GenBank accession numbers for the sequences used in this alignment are NC002035 for Fny-CMV, D12538 for Y-CMV, AM183118 for RI-8-CMV, AJ276480 for Mf-CMV, HE971489 for OSA3-CMV, QBH72281 for AZ14-CMV, CBG76802 for KS44-CMV, D86330 for SD-CMV, CAJ65577 for PI-1-CMV, FN552601 for 30RS-CMV, BAD15371 for TN-CMV, AF416900 for LS-CMV, and Q66125 for Q-CMV.

**Figure 9 viruses-16-00676-f009:**
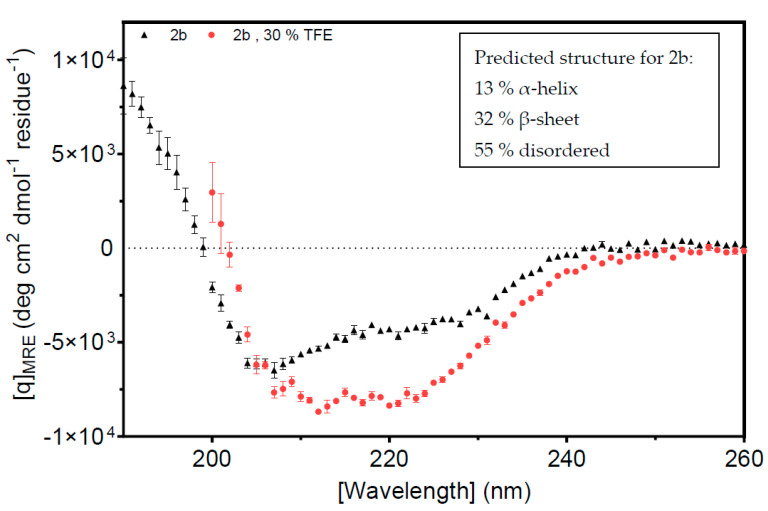
Circular dichroism analysis of the Fny-CMV 2b protein. Data were obtained at a protein concentration of 13.5 μM in 10 mM potassium phosphate buffer, pH 7.4, at 25 °C. The spectra were recorded between 190 nm and 240 nm, and the average of five scans is shown. The data were converted to mean residue ellipticity to correct for protein size and concentration, and the spectrum was run through the CDSSTR program to assess the average secondary structure composition. The Fny-CMV 2b protein (black) shows a mix of structural motifs, which, when fitted by K2D, show a mainly disordered protein (55%), with some α-helix (13%), and some β-strands (32%). Upon addition of 30% trifluoroethanol (TFE, red), there is a decrease in the amount of β-strand structure (14%) and an increase in the amount of α-helix (31%), with the random coil contingent remaining constant.

**Figure 10 viruses-16-00676-f010:**
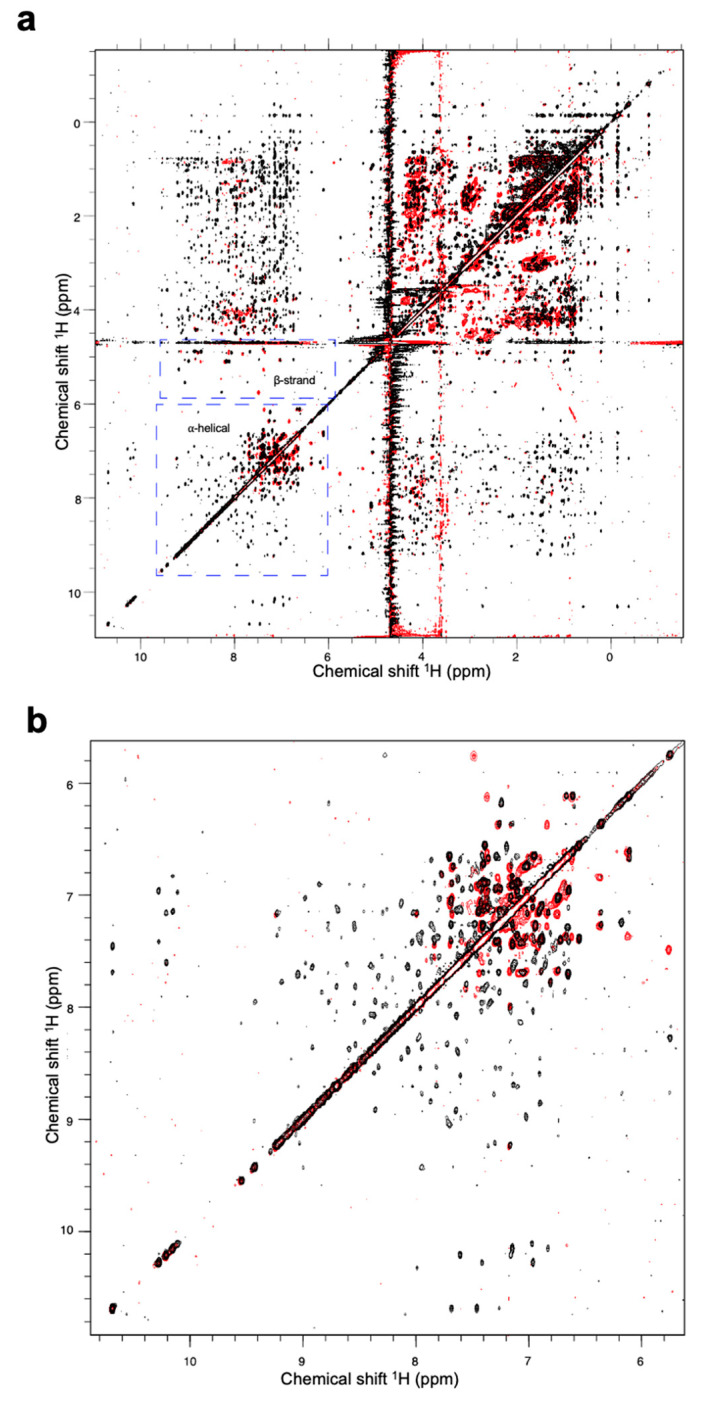
Proton nuclear magnetic resonance spectra of the Fny-CMV 2b protein. Results obtained for a 200 μM sample in phosphate-buffered saline (137 mM NaCl, 2.7 mM KCl, 10 mM Na_2_HPO_4_, 1.8 mM KH_2_PO_4_, pH 7.4). (**a**) Overlay of spectra from a 2D nuclear Overhauser effect spectroscopy (NOESY) experiment (800 MHz, 298K, mixing time = 150 ms) shown in black and a 2D total correlation spectroscopy (TOCSY) experiment (800 MHz, 298K, mixing time = 32 ms) shown in red. The results suggest the presence of approximately 40 residues with an α-helical secondary structure and approximately 20 residues with a β-sheet secondary structure. (**b**) An expansion of the region between 6 and 11 ppm containing information relating to aromatic side chains and amide protons.

## Data Availability

The data presented in this study are available in the article and Appendix A.

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
