# Peer review of "Investigating the Interactions of the Cucumber Mosaic Virus 2b Protein with the Viral 1a Replicase Component and the Cellular RNA Silencing Factor Argonaute 1"

_viruses, 2024, doi:10.3390/v16050676_

Round 1
Reviewer 1 Report
Comments and Suggestions for Authors
Crawshaw and colleagues describe a very interesting investigation into the factors affecting interactions between the CMV 2b protein and two other proteins - the virally-encoded 1a protein and the host-encoded AGO1 protein. This is a challenging topic, and their work makes important contributions to the field. The CMV 2b protein is a highly multifunctional counter-defence protein, and many of its reported functions require interaction with other host or viral proteins. The 2b protein is also reasonably small. As a consequence, there has been longstanding interest in how such a small protein interacts with so many different partners. These interactions also vary between the 2b proteins from different CMV strains, and influence differing viral symptomology. The authors focus specifically on the interactions between 2b, 1a and AGO1 because these three components all interact to establish a delicate balance between antiviral defence and transmission of CMV by its aphid vector. This is important both for understanding fundamental CMV biology and also for development of new approaches to crop virus control. Despite this importance, the interactions are not understood in detail. The authors initially focus on a mutational analysis of the 2b protein, deleting and substituting various regions of 2b then examining interactions with 1a or AGO1 by application of BIFC and co-immunoprecipitation. The selection of mutations is very comprehensive, drawing together existing evidence from several different studies so that effects can be directly compared. The balance of evidence from these careful examinations is that interactions between 2b and 1a or AGO1 likely cannot be explained by the effects of single/short regions of sequence. This leads to the part of the manuscript that I find most interesting - the authors examine the possibility that interactions may instead be driven by intrinsically disordered domains. Such domains, generally, can enable one protein to bind with many different partners, by acquiring structure and remodelling dependent upon the environmental conditions that protein is in. As the authors note, this is an appealing hypothesis that might explain how such a small protein can have such broad range of partners or effects. The authors use computational prediction to examine the likelihood that 2b has disordered domains, finding that this is indeed likely. They then use two different wet lab methods to test the computational predictions. The outcomes of one lab approach confirm computational predictions, detecting the presence of disordered domains. However, a second lab method did not detect disordered domains. The authors discuss clearly and reasonably possible explanations for this discrepancy. They also propose future work that may resolve the question.
Overall, this manuscript is written clearly and describes the experiments in details. Both the findings and the material generated will be an important resource for other CMV researchers. It also raises some thought provoking ideas. In sum, I support it's acceptance for publication. I have only one minor point I would appreciate the authors addressing: on ln 403-7, does the missing sequence identity between TAV 2b and CMV 2b correspond to regions where greater disorder are predicted?
Reviewer 2 Report
Comments and Suggestions for Authors
The manuscript by Crawshaw and colleagues describes the results of CMV 2b protein mutagenesis characterizing subcellular localization of the obtained mutant variants and their functional activity, in particular their ability to interact with AGO1 and 1a proteins. Authors tested both previously described 2b mutants and generated novel variants including the chimeric Fny/LS variants. Based on the obtained results authors suggest a hypothesis that could explain 2b interactions with AGO1 and 1a. The manuscript is well-structured, easy to read and understand. It contains a big massive of results obtained using wide specter of methods.
However, I have some major concerns and some minor points to address.
First is microscopy: the quality of images, resolution and magnification.
Figure 2. Despite there is control combination (top left) it’s hard to identify there the P-bodies. Authors should present the image with higher resolution and magnification, moreover, it would be appreciated by the reader if the P-bodies are marked with an arrow or somehow else. Also authors should add an image of 1a- and 2b- fusions per se (not in a combination) as an additional control. Despite for 2b such control is present as a panel of Fig. S3, it is hard to interpret because of low resolution.
Why different fusions are used in this experiment? According to my experience, sometimes fusions with RFP are distributed in the cell not in the same way as GFP-fused protein (even if the protein of interest is the same) and stability/integrity of such fuses could also be different.
As for interpretation of fluorescent microscopy results – the integrity of fusion proteins should be definitely taken into account because we see RFP/GFP signal but we are not actually sure if this signal really comes from the fusion protein or just from the degraded due to proteolysis variant of this fusion. The integrity and the level of each mutant variant accumulation should be assessed by western-blot. Actually, some demanded western-blots are presented in frame of the co-immunoprecipitation experiments. But they definitely indicate that mutant variants are accumulated differently than WT 2b fusion protein (multiple bands stained with anti-GFP antibodies are present). I assume that the efficiency of their synthesis is different as well.
Moreover, authors claim that “mutant versions of the 2b protein lacking residues at their C-terminus (such as 85-110, 74-110, 56-110 or 44-110) showed an increased nuclear localisation compared to the wild-type protein” (Fig. S3). For me it is not obvious at all from the presented images. To compare distribution in this case the fluorescent signal should be quantified. And as I’ve already mentioned, the level of expression of each mutant should be assessed and taken into account.
Figures 3 and 6 show the results obtained in BiFC system. However, they are hard to interpret because the cell borders are not visible, moreover, staining of plasma membrane or nucleus, or even merge with an image in visible light would be very helpful.
Also the images in panels of Fig 2, Fig 3 and Fig S10 are hard to compare with other panels of the same figure because of different magnification.
Figure 5. Here again panel with control combination demonstrates 2b-GFP+AGO1-RFP while other panels contain 2b(mut)-RFP and AGO1-GFP images. Also authors should add an image of AGO1 fusion per se (not in a combination) as an additional control. And for example, AGO1-GFP + RFP. This control could be important with regard to fusion proteins stability. Because free FP per se is usually distributed between nucleus and cytoplasm.
Minor points:
- Figure 4 and 7 – fusion proteins are designated as RFP-2b etc. Should it be 2b-RFP etc? The line with 1a-GFP (Fig 4) and AGO1-GFP (Fig 7) doesn’t correspond to +/- It should be shifted down a little bit
- I suggest that authors divide two panels of Figure 1 into two separate figures – one for Introduction section and the other would go to the Results section
